# Impaired Regulation of Emotion in Bipolar I Disorder: Behavioral and Neurophysiological Signatures

**DOI:** 10.3390/neurosci6010020

**Published:** 2025-03-03

**Authors:** Mathilde Carminati, Frédéric Isel, Josselin Houenou, Michèle Wessa, Chantal Henry

**Affiliations:** 1Département de Psychiatrie et de Médecine Addictologique, Hôpital Lariboisière-Fernand Widal, 75010 Paris, France; mathilde.carminati@gmail.com; 2Laboratoire Vision Action Cognition—EA 7326, Institut de Psychologie, Université Paris Cité, 92100 Boulogne-Billancourt, France; 3Laboratoire de Phonétique et Phonologie—UMR UMR7018, CNRS, Institut de Linguistique et Phonétique Générales et Appliquées (ILPGA), Université Sorbonne Nouvelle, 75005 Paris, France; 4NeuroSpin, CEA, Institut des Sciences du Vivant Frédéric Joliot, 91191 Gif-sur-Yvette, France; josselin.houenou@aphp.fr; 5Pôle de Psychiatrie, AP-HP, Faculté de Médecine de Créteil, DHU PePsy, Hôpitaux Universitaires Mondor, 94000 Créteil, France; 6Institut für Psychologie, Johannes Gutenberg-Universität Mainz, 55122 Mainz, Germany; michele.wessa@zi-mannheim.de; 7Département Santé, Université Paris Cité, 75006 Paris, France; ch.henry@ghu-paris.fr; 8Departement of Psychiatry, Service Hospitalo-Universitaire, GHU Paris Psychiatrie & Neurosciences, 75014 Paris, France

**Keywords:** bipolar disorder, emotion regulation, electroencephalography

## Abstract

People with bipolar disorder (BD) present with mood instability resulting from more frequent and intense emotions in response to environmental conditions relative to healthy subjects. The aim of this study was to investigate the time course of emotion regulation strategies, distraction, and reappraisal in euthymic BD patients (i.e., normal mood range) using electroencephalography (EEG). Fourteen BD patients and 13 matched healthy controls took part in an experiment constituting three conditions, i.e., a passive viewing of positive, negative, and neutral pictures, and two regulation conditions, one with a reappraisal strategy and the other with a distraction strategy. Critically, the ERP results indicated that during passive viewing, the Late Positive Potential (LPP) was larger in BD patients compared with healthy controls, but only for neutral pictures. During emotion regulation, LPP amplitude was reduced in distraction conditions compared with viewing ones, especially for negative emotions in both patients and controls. Importantly, LPP was reduced in reappraisal conditions compared with passive viewing in an early time window for negative emotions and in a later time window for positive emotions in controls but not in patients. Our findings showed that the temporal dynamics of emotion regulation by reappraisal are faster for negative than for positive emotions in controls but not in BD patients.

## 1. Introduction

Bipolar disorder (BD) is a frequent disorder with a high suicide rate and an impact on patients’ functioning [1,2]. It is defined by the recurrence of depressive and manic or mixed episodes according to the type of disorder. Epidemiologically, the global prevalence of bipolar disorder is 2.4%, with similar prevalence rates in the United States, Europe and Asia [1]. In France, where this study was conducted, the prevalence of bipolar disorder is estimated to be between 1% and 2.5% in general population studies (approximately 1,600,000 people). This prevalence is undoubtedly greatly underestimated [3]. It is estimated that in 30% of patients, the onset of the disease occurs during adolescence, between the ages of 18 and 24. In fact, between 50% and 66% of adults with bipolar disorder report that their symptoms began before the age of 19. Men and women are equally affected.

For type I bipolar disorder (presence of at least one manic or mixed episode), the prevalence is around 0.6% [1]. For type II bipolar disorder (presence of at least one hypomanic episode associated with at least one major depressive episode), the prevalence is around 0.4% according to DSM-IV-TR criteria [4].

One of the most impacted domains in BD is emotional processing, whose disruptions underlie many symptoms. Even during remitted phases, individuals with BD present with mood instability resulting from more frequent and intense emotions in response to environmental conditions relative to healthy subjects, which leads to mood instability [5]. Emotional hyper-reactivity and mood instability have a detrimental impact on functioning, relapses, and suicide attempts [6,7]. Individuals with BD use maladaptive forms of emotion regulation strategies, such as rumination and dampening, compared with controls [8,9]. Psychological models of BD emphasize emotion regulatory processes. For example, Response Styles Theory (RST) [10] proposes that responding to low mood by ruminating leads to further depression, whereas distraction upregulates mood [11]. Ruminating on positive affect would be expected to amplify high moods [12], whereas responding by dampening positive affect is potentially linked to depression [12].

The current physiological consensual model of BD assumes that a dysfunction in prefrontal cortex regulatory processes of the limbic structures is the main cause of the disease [13,14]. In the context of emotional regulation (ER), this study aligns with the theoretical framework proposed by [15]. Emotional regulation can be defined as the processes by which an individual exerts influence over the emotions they experience, as well as the way they experience and express these emotions. Gross’s theoretical framework [15] identifies two broad categories of strategies for modifying emotional feelings. The first class of strategies pertains to the antecedents of the emotional response, involving the modification of the information input to emotional processing prior to the generation of the emotional response. The second class of strategies involves the modification of at least one of the three components of the emotional response—expressive, cognitive or physiological—after its generation.

ER may be automatic or controlled. Two well-studied controlled or voluntary emotion regulation strategies are attentional distraction and cognitive reappraisal [16,17,18,19]. In distraction, an external cognitive task is used to take the attention away from a specific emotional stimulus. Reappraisal consists of reinterpreting a situation to find a new cognitive way to modify a bad mood or a poor mood into a more stable, confident and optimistic mood state. Several studies on emotions reported attenuated self-reported negative or positive affects after distraction and reappraisal [17,20,21].

Temporal characteristics of these two emotion regulation strategies in people with BD might be provided by electroencephalography (EEG). Current EEG studies have assessed the time course of emotion regulation processes in healthy participants through cognitive reappraisal [22,23,24] and distraction [25]. These studies have focused more specifically on a late event-related potential (ERP), called the late positive potential (LPP). The LPP is a positive slow modulation of the ERP with a posterior midline scalp distribution and an onset around 250 msec after stimulus presentation. In studies on emotions, the LPP is usually thought to mark both emotional reactivity and regulation. Importantly, the LPP is highly sensitive to the emotional intensity of stimuli and is larger for both pleasant and unpleasant stimuli than for neutral stimuli [26,27]. Both ER strategies were successful in reducing LPP magnitudes to emotional stimuli. Specifically, the LPP is reliably smaller when an unpleasant stimulus is cognitively evaluated in a neutral compared to a negative manner [22,23]. Distraction began modulating the LPP from its very beginning whereas reappraisal began modulating the LPP later [18,25]. To our knowledge, no study has researched the time course of ER strategies in individuals with BD.

The aim of this EEG study was to investigate the temporal dynamics of two emotion regulation strategies, distraction and reappraisal, in euthymic patients with bipolar disorder. For that purpose, we used an established experimental paradigm [18,20]. The euthymic phase in bipolar disorder is defined as a normal mood range, implying the absence of depressed or elevated mood. Symptoms in the euthymic phase are not entirely absent but are subdued enough, so that mood and normal activity are not largely affected [28].

We hypothesized exaggerated emotional responses in BD patients in self-reports and a larger amplitude of the LPP in passive viewing compared with healthy controls. Similarly, emotion regulation deficits are expected, reflected in the lower attenuation of the LPP in emotion regulation conditions.

## 2. Materials and Methods

### 2.1. Participants

Patients with bipolar disorder were recruited by two psychiatrists at the Expert Centre for Bipolar Disorders at the Albert Chenevier hospital in Créteil (France). The study was described in detail to the patients, and informed consent was obtained from all subjects involved in the study. Fourteen patients with bipolar I disorder (seven men and seven women; mean age = 39.5 years, SD = 11.06) and 14 volunteers took part in this experiment. Data from one participant in the control group were excluded from the analyses because it was extremely noisy. The final sample of the control group therefore consisted of 13 participants (seven men and six women; mean age = 28.5 years, SD = 12.7). The 27 participants were native French speakers, and their social and economic status (SES) was controlled.

A socio-demographic questionnaire was completed by all participants. Then, to assess mood state and executive functions—which may be implicated in emotional regulation—different scales and tests were administered to participants with bipolar disorder and controls: (1) MADRS depression scale [29]; (2) Young Mania Rating Scale (YMRS) [30]; (3) Affective Intensity Measure (AIM; Larsen, 1984) [31]; (4) Affective Liability Scale (ALS) [32]; (5) Multidimensional Assessment of Thymic State (MAThyS) [33]; (6) Emotion Regulation Questionnaire (ERQ) [34]; (7) Stroop test (current form: [35]); and (8) Trail Making Test (TMT) [36].

The 14 patients with bipolar disorder were all in the normothymic phase (i.e., not showing any thymic symptoms, depression or mania) at the time of the experimental protocol. To be considered normothymic, patients must not have had an acute episode in the previous three months (minimum), and they had to present stable scores on the depression scales (score ≤ 8 on the MADRS) and mania (score ≤ 8 on the YMRS). Thirteen patients were treated with a mood stabilizer and one patient with an antidepressant.

The patients were assessed at the Expert Centre for Bipolar Disorders at the Albert Chenevier Hospital in Créteil (France) and the diagnosis was made by one psychiatrist from this department. Patients were not included if they had:-current addictive comorbidities (use of alcohol, cannabis or other illicit substances),-a history of head trauma (with loss of consciousness), neurological pathologies (e.g., epilepsy, migraines),-a severe chronic pathology or if they were under legal protection (guardianship, curatorship, justice protection).

Healthy controls had no history of Axis I psychiatric disorders. Subjects with a neurological history were excluded. All participants were right-handed according to a handedness questionnaire [37], had normal or corrected-to-normal vision.

The study was conducted in accordance with the Declaration of Helsinki and approved by the Ethics Committee of Paris Descartes University (N° IRB: 20152900001072, 2015.

### 2.2. Stimulus Material

In this study we used the same material as in the Schönfelder et al. study [18]. The emotion-inducing stimuli were 90 color photographs drawn from the International Affective Picture System (IAPS) [38]. On the 9-point Self-Assessment Manikin scale (SAM) (5 being neutral) [39], negative and positive stimuli were highly arousing [on average, 6.1 (0.7)], and neutral stimuli were rated low in arousal [on average, 3.5 (0.1)]. Thus, 30 positive pictures (for example, exciting sport or happy family scenes) [averaged valence: 7.4 (0.3); averaged arousal: 6.2 (0.6)], 30 negative pictures (human violence, accidents) [averaged valence: 1.9 (0.3); averaged arousal: 6 (0.7)] and 30 neutral pictures (for example, people doing ordinary activities) [averaged valence: 5 (0.4); averaged arousal: 3.5 (0.1)] were selected. In order to ascertain the suitability of the 90 IAPS pictures selected for use within the French context, it would have been preferable to utilize the validation of 120 images of the IAPS in a French population aged from 20 to 88 years [40] (for a systematic review of the International Affective Picture System (IAPS) around the world, see [41]). However, in the interest of methodological comparability between studies, we chose to use the 90 IAPS pictures from the study by Schönfelder et al. [18].

For distraction, participants had to solve 3-operand arithmetic equations, including one subtraction and one addition (e.g., 8 + 3 − 7 = 5) and to indicate via button press as quickly and accurately as possible whether the displayed solution was correct or incorrect. Half of all equations were incorrect and were constructed to differ by 1 from the correct answer (Figure 1).

### 2.3. Experimental Task

The experiment consisted of three experimental conditions, a passive viewing condition and two regulation conditions, one with a reappraisal strategy and the other with a distraction strategy (Figure 1). The purpose of the passive viewing block was to establish the basic effect of emotional reactivity on the LPP. In reappraisal conditions, participants were given reappraisal instructions. The experimenter explained the concept of reappraisal as reinterpreting an unpleasant or a pleasant picture to decrease emotional impact. In distraction conditions, participants were asked to complete an arithmetic task. Before each picture presentation, a single-word instruction (VIEW, CALCULATE or REGULATE) was presented, signaling the strategy to be used during the following trial.

During passive viewing (VIEW), participants were asked to view the picture attentively and without trying to change upcoming emotional responses.

During distraction (CALCULATE), participants were asked to solve the concurrently presented mathematical equation and to indicate via button press whether the displayed solution was correct or incorrect whilst ignoring the background picture.

During reappraisal (REGULATE), participants were asked to cognitively diminish their emotional reactions by distancing themselves from the picture, by becoming a detached, uninvolved observer, or by thinking that the depicted situation was not real. Each picture was presented in the VIEW, CALCULATE and REGULATE condition, except for the neutral pictures that were not presented for reappraisal so as to not confuse participants by asking them to lower an emotional response to a non-affective stimulus. Each participant received a different pseudo-randomized trial order with no more than three trials of the same valence category or regulation instruction appearing consecutively. To practice the ER strategies, a train block was proposed before the experiment. Participants’ responses were reviewed by the experimenters until they felt sure that the ER instructions were correctly understood.

The time course of a trial was as follows: a fixation cross (500 ms) was presented on the center of the screen, followed by an instruction cue (2000 ms) that signaled to the participants to regulate their emotions according to the practiced strategies or to simply watch the picture, which was then replaced by the picture (5000 ms) (see Figure 1). For distraction trials, arithmetic problems were additionally presented as a transparent overlay on the picture to allow for a solution of the problem. After picture offset, participants rated their current emotional experience on a 9-point scale using the Self-Assessment Manikin scale for valence [39] ranging from pleasant (1) via neutral (5) to unpleasant (9). A variable inter-trial interval (3500–5500 ms) was presented prior to presentation of the next trial to permit recovery on all physiological measures. A total of five experimental blocks, separated by a brief rest, were administered. The complete experiment consisted of 240 trials and lasted about 65 min.

### 2.4. Electrophysiological Recordings

EEG was recorded from 64 electrodes, placed according to the international 10–20 system mounted in an elastic cap (ActiCap, Brain Products, Inning am Ammersee, Germany) and recorded with the Brain Vision Recorder, Brain Products. All channels were referenced online against FCz. Eye movement and blink artifacts were monitored with an electrooculogram (EOG). Vertical EOG was recorded bipolarly from electrodes placed above and below the right eye. Horizontal EOG was recorded bipolarly from positions at the outer canthus of each eye. For data analysis, channels were re-referenced to average electrodes. Electrode impedances were kept below 25 kΩ. Data were recorded at a sampling rate of 1000 Hz. An online band-pass filter of 0.01–100 Hz was used.

The continuous EEG data were filtered offline using a filter (IIR Butterworth filter) between 1 Hz (slope 24 dB/octave) and 40 Hz (slope 24 dB/octave). All recordings below 1 Hz or above 40 Hz were removed. On the continuous EEG data, automatic artifact detection for non-ocular artifacts was conducted (maximum amplitude difference in interval of 200 ms: 300 μV, maximum gradient voltage step: 70 μV/ms, lowest allowed activity in interval of 100 ms: 0.5 μV). Furthermore, we performed an ICA (Independent Component Analysis) in EEGLAB to identify blink components. Components identified as blinks [42] or muscle movements (probability > 0.9) were removed from the data.

The continuous data were epoched at −250 to 1200 msec relative to stimulus onset and were baseline-corrected relative to the average of the prestimulus interval (−250 to 0 msec). Grand-average ERPs were finally generated by computing the mean ERPs across participants in each condition. Data processing was performed offline using BrainVision Analyzer software version 2.0 (Brain Products, Inning am Ammersee, Germany).

Based on previous studies [18,22], LPP was quantified as the mean level of activity at an electrode cluster consisting of CP1, CPz, CP2, and Pz for the entire picture duration with the following time windows: 600–800 ms, 1000–2000 ms, 2000–3000 ms, and 3000–4000 ms et 4000–5000 ms.

### 2.5. Data Analysis

Subjective emotional state ratings and ERP were analyzed with Stastitica. Behavioral data and ERP responses were analyzed to evaluate emotional reactivity during free-viewing trials. Repeated-measures ANOVAs were performed with within-subject factor Emotion (positive, negative, neutral) and between-subject factor Group (patients with bipolar disorder, healthy controls). To evaluate regulation effects, we performed repeated-measures ANOVAs with Condition (view, distraction, reappraisal), Emotion (positive, negative) as within-subject factors and Group (healthy controls, patients with bipolar disorder) as the between-subject factor.

All ANOVA results were Greenhouse–Geisser corrected if the assumption of sphericity was violated. Effects with a significance level of <0.05 were treated as statistically significant. Post-hoc multiple comparisons were carried out using Bonferroni-adjusted corrections.

## 3. Results

### 3.1. Effect of Emotional Induction

#### 3.1.1. Subjective Assessment

Passive view

The ANOVA showed a significant effect of Emotion (*F*(2,50) = 109.72; *p* < 0.001; *η*^2^ = 0.81). Negative and positive trials differed from each other and from neutral trials indicating successful emotion induction. Emotional pictures (positive and negative) were experienced as more pleasant or more aversive than neutral pictures (all *p* < 0.001). (Figure 2). There were no group effects regarding BD patients and controls (*p* > 0.05).

Subjective assessment of emotional regulation

ANOVA revealed a significant Condition × Emotion interaction (*F*(2,50) = 23.32; *p* < 0.001; *η*^2^ = 0.48). Post-hoc analyses revealed that emotional pictures were rated as less negative or positive during distraction and reappraisal as compared with the free-viewing condition, this effect being larger for negative pictures (all *p* < 0.001). No difference was found between distraction and reappraisal conditions (*p* > 0.05). There were no group effects regarding BD patients and controls (*p* > 0.05) (Figure 3).

#### 3.1.2. ERP Responses on Passive View

On the 600–800 ms time window, the ANOVA revealed a significant Group × Emotion interaction (*F*(2,50) = 10.25; *p* < 0.001, *MSe* = 2.66; *η*^2^ = 0.29). Post-hoc analyses indicated that for neutral pictures the LPP amplitude was higher in patients with bipolar disorder than in healthy controls (*F*(1,25) = 5.93; *p* < 0.05; *MSe* = 2.57). No group difference was found for positive and negative pictures (*p* > 0.05). (Figure 4)

#### 3.1.3. ERP Responses in Emotional Regulation

On the 600–800 ms time window, the ANOVA revealed a significant Condition × Emotion (*F*(2,50) = 3.62; *p* < 0.05; *MSe* = 0.74, *η*^2^ = 0.13). Planned comparisons showed greater LPP amplitude for negative pictures in free-viewing conditions than in distraction conditions (*F*(1,25) = 5.15; *p* < 0.05; *MSe* = 1.75) meaning that distraction was able to reduce the response intensity. However, there were no effects in terms of the group regarding BD patients and controls.

Moreover, in reappraisal conditions, for the control participants, LPP amplitude was reduced compared to the free-viewing condition, in an early time window for negative emotions and in a later time window for positive emotions. This effect was not observed in patients with bipolar disorder, neither for positive nor for negative emotions.

## 4. Discussion

The aim of this study was to investigate behavioral and electrophysiological markers of two emotion regulation strategies, distraction and reappraisal, in euthymic patients with bipolar disorder. Patients were asked to report subjective assessments while an EEG was being recorded. The subjective evaluations showed no differences between healthy controls and patients with bipolar disorder as shown in previous studies [17,18,43], but we found some electrophysiological differences that are interesting to discuss. Behavioral results showed that both patients and controls experienced less intense emotions in response to positive and negative emotion-inducing IAPS pictures in both the distraction and reappraisal conditions compared with passive viewing.

At the neurophysiological level, during passive view, the LPP, i.e., an ERP component thought to reflect regulation processes, was larger for neutral pictures but not for pictures with an emotional valence (positive and negative) in patients with bipolar disorder compared with healthy controls. This is in favor of less regulation of emotional processes when the emotional signal is not clearly determined. Patients with bipolar disorder during the euthymic period failed to mobilize emotional regulatory processes when facing neutral stimuli, whereas there was no difference in comparison to control subjects when regulating emotional processes in front of stimuli with a clear emotional valence. These results are consistent with previous studies showing emotional hyperarousal in euthymic patients with bipolar disorder compared to control subjects when facing neutral stimuli. In contrast with the M’Bailara study [44], we did not find this result using qualitative behavioral measures (Likert scales). This can be explained by the fact that our sample size is much smaller than the one in M’Bailara’s study, where the number of subjects was 90 for the control group and 55 for the group of individuals with bipolar disorder, which seems to indicate that subjective tests are less sensitive in detecting this phenomenon than an electrophysiological marker [44,45,46]. Our results are in line with previous studies showing discrepancies between behavioral and neural activation measures in euthymic patients [17,47,48,49]. These results confirmed that failure of emotional regulation is a predominant feature of bipolar disorder [50,51]. It would be interesting to see if the dysregulation of emotional processes to neutral stimuli in euthymic patients with bipolar disorder extends to other types of stimuli (positive and negative) depending on the thymic episodes.

Regarding ERP measures, LPP amplitude was reduced in distraction conditions compared with free-viewing, especially for negative emotions in patients with BD as well as in controls. This is concordant with previous findings by [17], who did not find any differences between patients and controls in distraction conditions, meaning that patients can regulate emotional responses as well as controls.

Concerning the method of regulation by reappraisal, our study highlights a difference of temporality in the activation of emotional processes according to the valence of the images, between the control subjects and the patients with bipolar disorder. We found that LPP were reduced compared with free-viewing in controls but not in individuals with BD in an early time window for negative emotions and in a later time window for positive emotions. The controls thus seem to have a faster regulation process for negative stimuli than for positive stimuli, whereas the patients do not have a differentiated process depending on the valence. It may be clinically useful to train people with bipolar disorder to recognize and regulate negative emotions using stimuli from different modalities in context (face, emotional prosody, verbal) that could be presented in a serious game. Future studies in digital medicine could address this issue.

These results are to be related to the results of imaging studies showing overall abnormalities in the emotion regulation system in people with bipolar disorder. In [17], individuals with BD showed a pattern of limbic hyperactivation and altered connectivity with the frontal regions, resulting in an emotion regulation deficit with respect to amygdala downregulation during the reappraisal condition, mediated by reduced altered connectivity between the OFC and the amygdala. This regulation deficit was only present in reappraisal conditions and not during distraction. Moreover, reappraisal deficits are also found in the population at risk of bipolar disorder [17], and in individuals with schizophrenia [52]. Another study comparing both strategies [53] found that people with BD showed aberrantly increased intra-network and inter-network connectivity in the default mode network during distraction compared to the reappraisal condition, which indicates that these strategies can be distinguished by specific activity patterns in largescale brain networks.

## 5. Limitations

Certain limitations should be considered. First, the main limitation of the present study is the sample size, resulting in reduced power. It may be necessary to further replicate the study with a larger sample size. Additionally, reappraisal is more difficult to operationalize than distraction. Participants may not always succeed in implementing this strategy during the experiment. A recent study highlighted the fact that individuals prefer using distraction in the presence of high-intensity stimuli while they prefer using reappraisal in the case of low-intensity emotional stimuli [54].

However, our results seem to confirm that distraction may be a resource that individuals with BD can exploit during euthymic phases, while they may not benefit from applying reappraisal [55]. Further neuroimaging studies should be conducted to compare a range of emotion regulation strategies [56] in individuals diagnosed with different types of bipolar disorder. Moreover, future neurocognitive investigations should be conducted to better understand the role of prefrontal cortex networks [57,58,59] in emotion dysregulation.

## Figures and Tables

**Figure 1 neurosci-06-00020-f001:**
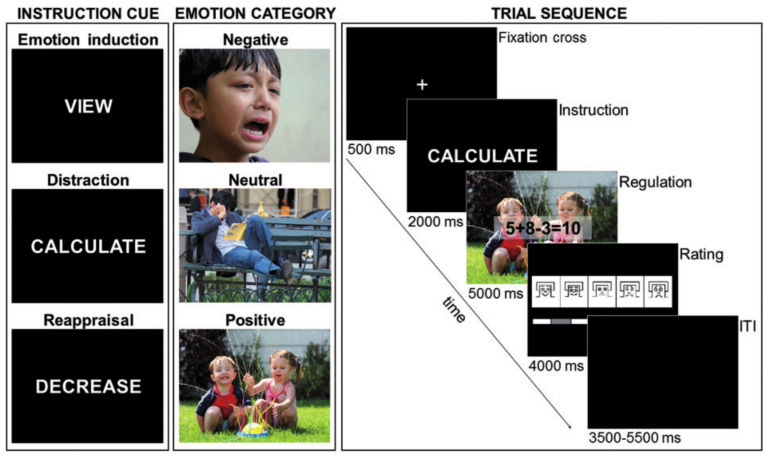
Illustration of the different experimental conditions (view, distraction and reappraisal) and trial sequence. The image shown in Figure 1 is taken from the International Affective Picture System (IAPS) [38]. Reprinted with permission from Schönfelder et al. study [18], p. 1312.

**Figure 2 neurosci-06-00020-f002:**
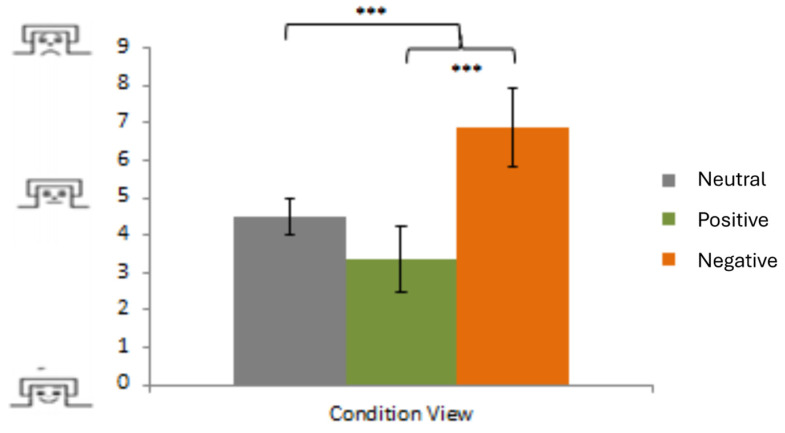
Behavioral measures of emotional response (mean emotion experience ratings) on a 9-point scale using the Self-Assessment Manikin scale for valence ranging from pleasant (1) via neutral (5) to unpleasant (9) during free-viewing trials; *** *p* < 0.001.

**Figure 3 neurosci-06-00020-f003:**
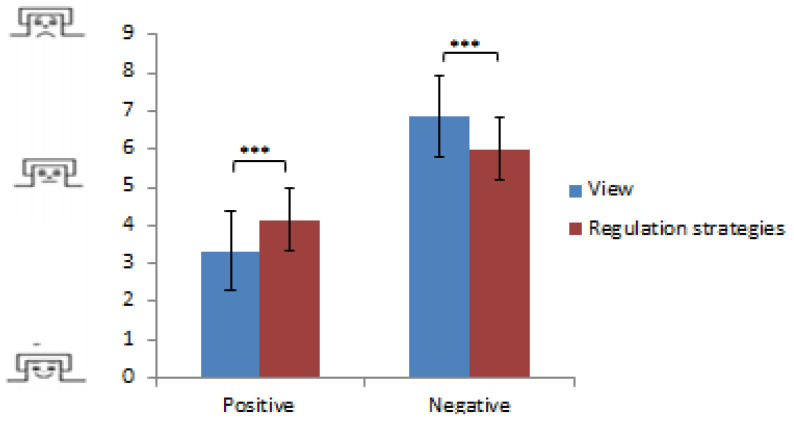
Behavioral measures of emotional response (mean emotion experience ratings) on a 9-point scale using the Self-Assessment Manikin scale for valence ranging from pleasant (1) via neutral (5) to unpleasant (9) during free-viewing and regulation (distraction and reappraisal) trials for positive and negative pictures; *** *p* < 0.001.

**Figure 4 neurosci-06-00020-f004:**
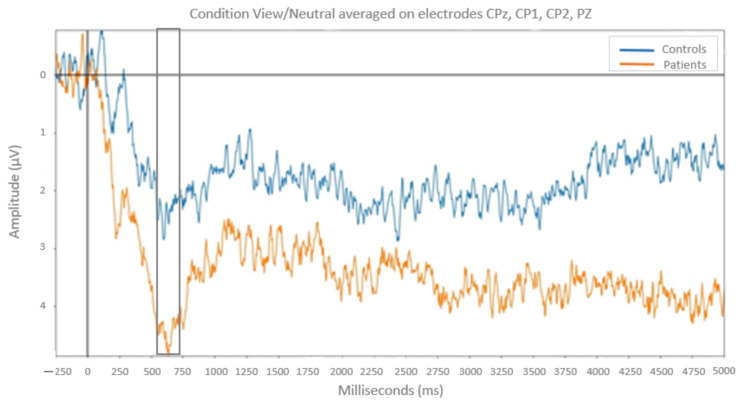
Effect of the presentation of neutral pictures on evoked potentials in the free-viewing condition, in patients (orange) and in healthy controls (blue). The gray box symbolizes the time window corresponding to the analysis (600–800 ms).

## Data Availability

Data are unavailable due to privacy restrictions.

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
