# Peer review of "Impaired Regulation of Emotion in Bipolar I Disorder: Behavioral and Neurophysiological Signatures"

_neurosci, 2025, doi:10.3390/neurosci6010020_

Round 1

Reviewer 1 Report

Comments and Suggestions for Authors

This manuscripts examines impaired emotion regulation in patients with bipolar disorder though behavioral and neurophysiological measures base don EEG. The approach involves emotion regulation strategies such as distraction and reappraisal. in an experiment with passive viewing of images. Results suggest that BD patients had larger LPP amplitudes during passive viewing of neutral images, suggesting a failure to regulate emotions in response to neutral stimuli. When using emotion regulation strategies, both groups showed reduced LPP amplitudes in the distraction condition compared to passive viewing, particularly for negative emotions. However, in the reappraisal condition, LPP amplitude reduction occurred only in controls, not in BD patients, indicating a delayed or impaired regulation of emotional responses. This suggests that BD patients fail to regulate emotions as effectively as controls.

The results show differences between BD patients and healthy controls in the ability to regulate emotions, especially for neutral stimuli and the reappraisal strategy.

Main points:

The study should provide more details on how the valence and arousal of the images presented to participants were controlled: This is important for replication as these affective dimensions are crucial for interpreting emotional and neurophysiological responses. t-test can be conducted to control differences across groups. To improve replications, more info can be provided. E.g., number of images used can be also provided in an appendix.

Another point is related to the use of the IAPS. To my knowledge, considering that, according to a recent review, this tool is not so used in France. It would suggest the authors to discuss the suitability of IAPS within the French context and, if necessary. Please see:

https://www.mdpi.com/1424-8220/23/8/3866

Another point is related to sample sizes, which should be discussed. Why control group is smaller tan BD?

The study should also address potential intra-individual variables that need to be controlled, as these can significantly impact the results. Factors such as individual differences in baseline emotional reactivity, cognitive style, or prior experiences with emotional regulation strategies could influence how participants respond to the emotional stimuli and regulation tasks. For example, participants with different levels of emotional awareness or coping skills might process the images differently, affecting the neurophysiological outcomes, such as the LLP of interest.

Lastly, there is a lack of detailed information regarding how noise in the EEG data was treated. EEG recordings are highly susceptible to various forms of noise, including ocular artifacts, muscle movements, and environmental interference, all of which can distort the signals. The study mentions artifact detection but does not provide sufficient detail.

Comments on the Quality of English Language

I am not an english native speaker.

Author Response

For research article

Response to Reviewer #1 Comments

1. Summary

2. Questions for General Evaluation

Reviewer’s Evaluation

Response and Revisions

Does the introduction provide sufficient background and include all relevant references?

Yes

Thank you for this positive evaluation.

Are all the cited references relevant to the research?

Yes/Can be improved/Must be improved/Not applicable

We have added several references.

Is the research design appropriate?

Yes/Can be improved/Must be improved/Not applicable

We specified the research design, concerning the IAPS images and the individuals with BD

Are the methods adequately described?

Yes/Can be improved/Must be improved/Not applicable

We improved the description of the Materials and Methods section.

Are the results clearly presented?

Yes/Can be improved/Must be improved/Not applicable

We clarified some points.

Are the conclusions supported by the results?

Yes/Can be improved/Must be improved/Not applicable

We added information in the Discussion.

3. Point-by-point response to Comments and Suggestions for Authors

Comments 1: [The study should provide more details on how the valence and arousal of the images presented to participants were controlled: This is important for replication as these affective dimensions are crucial for interpreting emotional and neurophysiological responses. t-test can be conducted to control differences across groups. To improve replications, more info can be provided. E.g., number of images used can be also provided in an appendix.]

Response 1: Thank you for this useful comment. We agree with this comment. Therefore, we added in the revised manuscript the average values for valence and arousal of the three types of emotional pictures:

p.3, paragraph 1, lines 95-104: [In this study we used the same material as in Schönfelder et al. study [12]. Emotion-inducing stimuli were 90 color photographs drawn from the International Affective Picture System (IAPS) [25]. On the 9-point Self-Assessment Manikin scale (SAM) (5 being neutral) (Bradley and Lang (1994)), negative and positive stimuli were highly arousing [in average, 6.1 (0.7)], and neutral stimuli were rated low in arousal [in average, 3.5 (0.1)]. Thus, 30 positive pictures (for example, exciting sport or happy family scenes) [averaged valence: 7.4 (0.3); averaged arousal: 6.2 (0.6)], 30 negative pictures (human violence, accidents) [averaged valence: 1.9 (0.3); averaged arousal: 6 (0.7)] and 30 neutral pictures (for example, people doing ordinary activities) [averaged valence: 5 (0.4); averaged arousal: 3.5 (0.1)] were selected.]

Comments 2: [Another point is related to the use of the IAPS. To my knowledge, considering that, according to a recent review, this tool is not so used in France. It would suggest the authors to discuss the suitability of IAPS within the French context and, if necessary. Please see:

https://www.mdpi.com/1424-8220/23/8/3866]

Response 2: We agree with this comment, and we thank Reviewer #1 for mentioning the systematic review of IAPS around the world by Branco et al. (2023). We have added a paragraph on this subject on p. 3, paragraph 1, lines 104-110.

[In order to ascertain the suitability of the 90 IAPS pictures selected for use within the French context, it would have been preferable to utilize the validation of 120 images of the IAPS in a French population aged from 20 to 88 years, as conducted by Bungener, Bonnet, and Fiori-Duharcourt (2016) (for a systematic review of International Affective Picture System (IAPS) around the World, see Branco et al., 2023). However, in the interest of methodological comparability between studies, we chose to use the 90 IAPS pictures from the study by Schönfelder et al [12].]

Comments 3: [Another point is related to sample sizes, which should be discussed. Why control group is smaller than BD?]

Response 3: We thank Reviewer #1 for pointing out this methodological element. The control group consisted of one fewer participant than the bipolar group. This is because data from one control participant was excluded from the analyses because it was extremely noisy. We added this information in the Participants Section p. 2, paragraph 5, lines 82-86:

[Fourteen bipolar patients (seven men and seven women; mean age = 39.5 years, SD = 11.06) and fourteen volunteers took part in this experiment. Data from one participant in the control group was excluded from the analyses because it was extremely noisy. The final sample of the control group therefore consisted of 13 participants (seven men and six women; mean age = 28.5 years, SD = 12.7).]

Comments 4: [The study should also address potential intra-individual variables that need to be controlled, as these can significantly impact the results. Factors such as individual differences in baseline emotional reactivity, cognitive style, or prior experiences with emotional regulation strategies could influence how participants respond to the emotional stimuli and regulation tasks. For example, participants with different levels of emotional awareness or coping skills might process the images differently, affecting the neurophysiological outcomes, such as the LLP of interest.]

Response 4: We agree with this comment of the Reviewer. To emphasize this point, we added the following information in the Participants section pp. 2-3, 87-95:

[A socio-demographic questionnaire was completed by all participants. Then, to assess mood state and executive functions – which may be implicated in emotional regulation – different scales and tests were administered to bipolar participants and controls: (1) MADRS depression scale (Montgomery and Asberg, 1979); (2) Young Mania Rating Scale (YMRS) (Young et al., 1978); (3) Affective Intensity Measure (AIM; Larsen, 1984); (4) Affective Liability Scale (ALS; Harvey, Greenberg et Serper, 1989); (5) Multidimensional Assessment of Thymic State (MAThyS, Henry, M’Baïlara, Matthieu, Poinsot et Falissard, 2008); (6) Emotion Regulation Questionnaire (ERQ, Gross et John, 2003); (7) Stroop test (current form: Albaret & Migliore, 1999); (8) Trail Making Test (TMT; Marvin, 2012).]

Interestingly, the inter-individual variability indicated by the standard deviations on the various scales mentioned above varied little between the group of bipolar individuals and the control group. This suggests that our group of patients were relatively homogeneous.

We fully agree with Reviewer #1 that factors such as individual differences in baseline emotional reactivity, cognitive style, or prior experiences with emotional regulation strategies could influence how participants respond to the emotional stimuli and regulation tasks. In that sense, it would have been interesting to conduct correlational analyses using these different variables. However, given the size of our samples, we did not take the risk of conducting such analyses, the statistical validity of which would have been questionable. 

Comments 5: [Lastly, there is a lack of detailed information regarding how noise in the EEG data was treated. EEG recordings are highly susceptible to various forms of noise, including ocular artifacts, muscle movements, and environmental interference, all of which can distort the signals. The study mentions artifact detection but does not provide sufficient detail.]

Response 5: We thank the Reviewer for this comment. We have therefore revised section 2.4 ‘Electrophysiological recordings’ to give more details on how the EEG data were pre-processed p. 4, paragraph 3, lines 167-188.

[EEG was recorded from 64 electrodes, placed according to the international 10–20 system mounted in an elastic cap (ActiCap, Brain Products) and recorded with the Brain Vision Recorder, Brain Products.  All channels were referenced online against FCz. Eye movement and blink artifacts were monitored by the electrooculogram (EOG). Vertical EOG was recorded bipolar from electrodes placed above and below the right eye. Horizontal EOG was recorded bipolar from positions at the outer canthus of each eye. For data analysis, channels were re-referenced to average electrodes. Electrode impedances were kept below 25 kΩ. Data were recorded at a sampling rate of 1000 Hz. An online band-pass filter of 0.01– 100 Hz was used.

The continuous EEG data were filtered offline using a filter (IIR Butterworth filter) between 1 Hz (slope 24 dB/octave) and 40 Hz (slope 24 dB/octave). All recordings below 1 Hz or above 40 Hz were removed. On the continuous EEG data, automatic artifact detection for non-ocular artifacts was conducted (Maximum amplitude difference in interval of 200 ms: 300 μV, maximum gradient voltage step: 70 μV/ms, lowest allowed activity in interval of 100 ms: 0.5 μV). Furthermore, we performed an ICA (Independent Component Analysis) in EEGLAB to identify blink components. Components identified as blinks (see Debener et al., 2010) or muscle movements (probability >0.9) were removed from the data.

The continuous data were epoched at -250 to 1200 msec relative to stimulus onset and were baseline-corrected relative to the average of the prestimulus interval (-250 to 0 msec). Grand-average ERPs were finally generated by computing the mean ERPs across participants in each condition. Data processing was performed offline using BrainVision Analyzer 2 software (Brain Products).]

4. Response to Comments on the Quality of English Language

Reviewer #1 does not comment on the quality of English language, and she/he just mentioned “I am not an english native speaker.”

Reviewer 2 Report

Comments and Suggestions for Authors

The study idea is smart; please specify in the title that you were dealing with patient with bipolar I disorders, and not with the patient suffering from hypomania and depression disorders. (PB II).

Abstract: Please consider that more frequent intense emotions in response to environmental conditions is the core of mood instability, and as such, these more frequent intense emotions in response to environmental conditions, do not lead to mood instability, but they are actually the core of mood instability.

Please define what do you mean with euthymic BD patients. Further, please consider not to define a person by its disorder or illness; accordingly, please always talk about a person with bipolar disorder, and not a bipolar person or a bipolar patient. For the patient, please report more details, such as mean, age, generational, and the duration of bipolar disorder in this, along with the current state of bipolar disorder. For the letter, it means whether these individuals were in a manic state, in a new state, or in a depressed state. For the last sentence of the abstract, this is somehow surprising, in that the preview descriptions of the data did not suggest such a pattern of results. Please delete the expression of neurochronometry, as this expression is not standardized. Given this, I suggest to revise and improve the abstract.

Introduction: please report the exact prevalence rates of blood pressure I and II; please do not always talk about patients, because not all individuals suffering from a psychiatric illness are also automatically also patients; in this view, the expression patient implicitly denotes a person hospitalized in a clinic. For the sentence regarding emotional regulation strategies of individuals with bipolar disorder, please specify, if rumination and dampening occur in the manic phase, in the hypomanic phase, in the neutral phase or in the depressed phase. For the topic of emotional regulation, please report to which theoretical framework you refer to; more specifically, the emotional regulation concept of James Cross (Becerra et al., 2020; Gross, 2015; Gross, 1998, 2002; Gross, 2013, 2023; Gross and Barrett, 2011; Gross and John, 2003)does not distinguish between attention, destruction and cognitive, free, appraiser, but between cognitive free appraisal and emotion suppression. Further, please consider that cognitive reappraisal is not to reduce its emotional impact of an emotion or a stimulus, but the aim is to find a new cognitive way to modify bad mood or a poor mood into more stable and confident and optimistic mood state. Relatedly, please describe in more details. The following sentence: several studies. Further, please consider that cognitive reappraisal is not to reduce its emotional impact of an emotion or a stimulus, but the aim is to find a new cognitive way to modify bad mood or a poor mood into more stable and confident and optimistic mood state. Relatedly, please describe in more details. The following sentence: several studies reported at weight, self-reported negative or positive effect after distraction and reappraisal. Please consider that they are not ‘emotional studies’, but there are ‘studies on emotions’. I did highly appreciate that the authors formulated hypotheses.

Materials and methods: please describe in more details the characteristics of patients; more specifically, describe the duration of bipolar disorder, illness, including the number of both manic and depressive states. Further, please report patient medication intake, including the type and dosage of medication. Overall, the method section was particularly nicely crafted. Results: Overall, the method section was particularly nicely crafted. Results: this part was particularly nicely crafted. Discussion: this part was particularly nicely crafted. The authors might consider if into what extent their research results might be clinically useful in the treatment of individuals suffering from bipolar disorders.

References

Becerra, R., Preece, D.A., Gross, J.J., 2020. Assessing beliefs about emotions: Development and validation of the Emotion Beliefs Questionnaire. PloS one 15(4), e0231395.

Gross, J., 2015. Emotion Regulation: Current Status and Future Prospects. Psychological Inquiry 26, 1-26.

Gross, J.J., 1998. The Emerging Field of Emotion Regulation: An Integrative Review. Review of General Psychology 2(3), 271-299.

Gross, J.J., 2002. Emotion regulation: Affective, cognitive, and social consequences. Psychophysiology 39(3), 281-291.

Gross, J.J., 2013. Emotion regulation: taking stock and moving forward. Emotion 13(3), 359-365.

Gross, J.J., 2023. Conceptual foundations of emotion regulation, in: Gross, J.J.F., B., Q,; (Ed.) Handbook of Emotion Regulation Guilford Press, New York NY, USA.

Gross, J.J., Barrett, L.F., 2011. Emotion Generation and Emotion Regulation: One or Two Depends on Your Point of View. Emotion review : journal of the International Society for Research on Emotion 3(1), 8-16.

Gross, J.J., John, O.P., 2003. Individual differences in two emotion regulation processes: implications for affect, relationships, and well-being. J Pers Soc Psychol 85(2), 348-362.

Author Response

Response to Reviewer #2 Comments

1. Summary

2. Questions for General Evaluation

Reviewer’s Evaluation

Response and Revisions

Is the work a significant contribution to the field?

Thank you for this evaluation.

Is the work well organized and comprehensively described?

Thanks to the fruitful comments and suggestions of Reviewer #2, we have described certain aspects of the study in more detail. 

Is the work scientifically sound and not misleading?

Thank you.

Are there appropriate and adequate references to related and previous work? 

We added several references in the revised manuscript.

Is the English used correct and readable?        

We improved the English form.

3. Point-by-point response to Comments and Suggestions for Authors

Comments 1: [The study idea is smart; please specify in the title that you were dealing with patient with bipolar I disorders, and not with the patient suffering from hypomania and depression disorders. (PB II).]

Response 1: We thank Reviewer #1 for encouragement concerning our study. Thank you for pointing this out. We have replaced “bipolar disorder” by “bipolar I disorder” in the title page 1, line 2.

Comments 2: [Abstract: Please consider that more frequent intense emotions in response to environmental conditions is the core of mood instability, and as such, these more frequent intense emotions in response to environmental conditions, do not lead to mood instability, but they are actually the core of mood instability.]

Response 2: We agree. We have, accordingly reformulated the sentence in the abstract lines 19-20:

[Patients with bipolar disorder (BD) present with mood instability resulting from more frequent and intense emotions in response to environmental conditions relative to healthy subjects.]

Comments 3:

Point 1: [Please define what do you mean with euthymic BD patients.]

Response 1: Thank you for this suggestion. We specify the notion of “euthymic phase” in the abstract lines 22-23:

[in euthymic BD patients (i.e. normal range of mood)]

Moreover, we added the definition of euthymic phase and a reference (Bhatia, Sidana, and Bajaj, 2018) p. 2, paragraph 4, lines 73-76:

[Euthymic phase in BPAD is defined as normal range of mood, implying the absence of depressed or elevated mood. Symptoms in euthymic phase are not entirely absent but are subdued enough, so that mood and normal activity are not largely affected.]

Point 2: [Further, please consider not to define a person by its disorder or illness; accordingly, please always talk about a person with bipolar disorder, and not a bipolar person or a bipolar patient.]

Response 2: Thank you for pointing this out. We changed accordingly. 

Point 3: [For the patient, please report more details, such as mean, age, generational, and the duration of bipolar disorder in this, along with the current state of bipolar disorder.]

Response 3: We thank Reviewer #1 for this important comment, and we added information about patients in the Participants section, p. 2, last paragraph, lines 83-116:

[Patients with bipolar disorder were recruited by two psychiatrists at the Expert Centre for Bipolar Disorders at the Albert Chenevier hospital in Créteil (France). The study was described in detail to the patients, and informed consent was obtained from all subjects involved in the study. Fourteen patients with bipolar I disorder (seven men and seven women; mean age = 39.5 years, SD = 11.06) and fourteen volunteers took part in this experiment. Data from one participant in the control group was excluded from the analyses because it was extremely noisy. The final sample of the control group therefore consisted of 13 participants (seven men and six women; mean age = 28.5 years, SD = 12.7). The twenty-seven participants were native French speakers, and their social and economic status (SES) was controlled.

A socio-demographic questionnaire was completed by all participants. Then, to assess mood state and executive functions – which may be implicated in emotional regulation – different scales and tests were administered to participants with bipolar disorder and controls: (1) MADRS depression scale (Montgomery and Asberg, 1979); (2) Young Mania Rating Scale (YMRS) (Young et al., 1978); (3) Affective Intensity Measure (AIM; Larsen, 1984); (4) Affective Liability Scale (ALS; Harvey, Greenberg et Serper, 1989); (5) Multidimensional Assessment of Thymic State (MAThyS, Henry, M’Baïlara, Matthieu, Poinsot et Falissard, 2008); (6) Emotion Regulation Questionnaire (ERQ, Gross et John, 2003); (7) Stroop test (current form: Albaret & Migliore, 1999); (8) Trail Making Test (TMT; Marvin, 2012).

The 14 patients with bipolar disorder were all in the normothymic phase (i.e. not showing any thymic symptoms, depression or mania) at the time of the experimental protocol. To be considered normothymic, patients must not have had an acute episode in the previous three months (minimum), and they had to present stable scores on the depression scales (score ≤ 8 on the MADRS - Montgomery-Åsberg Depression Rating Scale, 1979) and mania (score ≤ 8 on the YMRS - Young Mania Rating Scale, Young, 1978).

The patients were assessed at the Expert Centre for Bipolar Disorders at the Albert Chenevier Hospital in Créteil (France) and the diagnosis was made by one psychiatrist from this department. Patients were not included if they had:

- current addictive comorbidities (use of alcohol, cannabis or other illicit substances),

- a history of head trauma (with loss of consciousness), neurological pathologies (e.g. epilepsy, migraines),

- a severe chronic pathology or if they were under legal protection (guardianship, curatorship, justice protection).]

Point 4: [For the letter, it means whether these individuals were in a manic state, in a new state, or in a depressed state.]

Response 4: We're sorry, but we don't understand what Reviewer #2 means by his remark about ‘the letter’?

Point 5: [For the last sentence of the abstract, this is somehow surprising, in that the preview descriptions of the data did not suggest such a pattern of results.]

Response 5: We agree with Reviewer #1 that this sentence was ambiguous considering the results. This conclusion only applies to regulation by reappraisal. We have therefore modified the summary accordingly lines 33-34.

Point 6: [Please delete the expression of neurochronometry, as this expression is not standardized.]

Response 6: We agree, and we replaced line 33 of the abstract “neurochronometry” by “temporal dynamics”.

Point 7: [Given this, I suggest to revise and improve the abstract.]

Response 7: We revised the abstract accordingly. We thank Reviewer #1 for the fruitful comments, which improved this part of the manuscript.

Point 8: [Introduction: please report the exact prevalence rates of blood pressure I and II.]

Response 8: Thank you for pointing this out. We added a paragraph in the Introduction section pp. 1-2, lines 42-53:

[Epidemiologically, the global prevalence of bipolar disorder is 2.4%, with similar prevalence rates in the United States, Europe and Asia (Merikangas et al., 2011). In France, the prevalence of bipolar disorder is estimated to be between 1% and 2.5% in general population studies (approximately 1,600,000 people). This prevalence is undoubtedly greatly underestimated (Hamon, 2010). It is estimated that in 30% of patients, the onset of the disease occurs during adolescence, between the ages of 18 and 24. In fact, between 50% and 66% of adults with bipolar disorder report that their symptoms began before the age of 19. Men and women are equally affected. For type I bipolar disorder (presence of at least one manic or mixed episode), the prevalence is around 0.6% (Merikangas et al., 2011). For type II bipolar disorder (presence of at least one hypomanic episode associated with at least one major depressive episode), the prevalence is around 0.4% according to DSM-IV-TR criteria.]

Point 9: [Please do not always talk about patients, because not all individuals suffering from a psychiatric illness are also automatically also patients; in this view, the expression patient implicitly denotes a person hospitalized in a clinic.]

Response 9: We agree, and we changed accordingly in the whole manuscript.

Point 10: [For the sentence regarding emotional regulation strategies of individuals with bipolar disorder, please specify, if rumination and dampening occur in the manic phase, in the hypomanic phase, in the neutral phase or in the depressed phase.]

Response 10: Thank you for this suggestion. We added a paragraph in the Introduction section, p. 2, paragraph 3, lines 61-66:

[Psychological models of BD emphasise emotion regulatory processes. For example, Response Styles Theory (RST; Nolen-Hoeksema, 1991) proposes that responding to low mood by ruminating leads to further depression, whereas distraction upregulates mood (Lyubomirsky et al, 2015). Ruminating on positive affect would be expected to amplify high moods (Feldman et al., 2008), whereas responding by dampening positive affect is potentially linked to depression (Feldman et al., 2008).]

Point 11: [For the topic of emotional regulation, please report to which theoretical framework you refer to; more specifically, the emotional regulation concept of James Cross (Becerra et al., 2020; Gross, 2015; Gross, 1998, 2002; Gross, 2013, 2023; Gross and Barrett, 2011; Gross and John, 2003)does not distinguish between attention, destruction and cognitive, free, appraiser, but between cognitive free appraisal and emotion suppression.]

Response 11: Thank you for this important comment. We added the information concerning the theoretical framework you refer to in the Introduction section p. 2, paragraph 4, lines 70-79:

[In the context of emotional regulation, this study aligns with the theoretical framework proposed by Gross (2008). Emotional regulation can be defined as the processes by which an individual exerts influence over the emotions they experience, as well as the way they experience and express these emotions. Gross's theoretical framework identifies two broad categories of strategies for modifying emotional feelings: The first class of strategies pertains to the antecedents of the emotional response, involving the modification of the information input to emotional processing prior to the generation of the emotional response. The second class of strategies involves the modification of at least one of the three components of the emotional response – expressive, cognitive or physiological – after its generation.]

Point 12: [Further, please consider that cognitive reappraisal is not to reduce its emotional impact of an emotion or a stimulus, but the aim is to find a new cognitive way to modify bad mood or a poor mood into more stable and confident and optimistic mood state.]

Response 12: Thank you for this fruitful precision. We modified a sentence in the Introduction section p. 2, paragraph 5, lines 83-85:

[Reappraisal consists of reinterpreting a situation to find a new cognitive way to modify bad mood or a poor mood into more stable and confident and optimistic mood state.]

Point 13: [The following sentence: several studies reported at weight, self-reported negative or positive effect after distraction and reappraisal. Please consider that they are not ‘emotional studies’, but there are ‘studies on emotions’. I did highly appreciate that the authors formulated hypotheses.]

Response 13: Thank you for this comment. We changed accordingly.

Point 14: [Materials and methods: please describe in more details the characteristics of patients; more specifically, describe the duration of bipolar disorder, illness, including the number of both manic and depressive states.]

Response 14: This was done in answer 3 above.

Point 15: [Further, please report patient medication intake, including the type and dosage of medication. Overall, the method section was particularly nicely crafted.]

Response 15: We thank Reviewer #2 to point this out.

The following sentence was added in Materials and Methods Section p.3, paragraph 3, lines 139-140:

[Thirteen patients were treated with a mood stabiliser and one patient with an antidepressant.]

Point 16: [Results: this part was particularly nicely crafted.]

Response 16: Thank you for this encouraging comment.

Point 17: [Discussion: this part was particularly nicely crafted. The authors might consider if into what extent their research results might be clinically useful in the treatment of individuals suffering from bipolar disorders.]

Response 17: Thank you for this important suggestion. We added a sentence in the Discussion Section p. 9, paragraph 3, lines 357-360:

[It may be clinically useful to train people with bipolar disorder to recognise and regulate negative emotions using stimuli from different modalities in context (face, emotional prosody, verbal) that could be presented in a serious game. Future studies in digital medicine could address this issue.]

4. Response to Comments on the Quality of English Language

Point 1: No points mentioned

Response 1:

Round 2

Reviewer 1 Report

Comments and Suggestions for Authors

Thank you for addressing my comments.